# Unbiased Quantitative Single-Cell Morphometric Analysis to Identify Microglia Reactivity in Developmental Brain Injury

**DOI:** 10.3390/life13040899

**Published:** 2023-03-28

**Authors:** Mark St. Pierre, Sarah Ann Duck, Michelle Nazareth, Camille Fung, Lauren L. Jantzie, Raul Chavez-Valdez

**Affiliations:** 1Division of Neonatal-Perinatal Medicine, Department of Pediatrics, Johns Hopkins School of Medicine, Baltimore, MD 21287, USA; 2Department of Molecular and Cellular Biology, Johns Hopkins University Krieger School of Arts and Sciences, Baltimore, MD 21205, USA; 3Division of Neonatology, Department of Pediatrics, University of Utah, Salt Lake City, UT 84132, USA

**Keywords:** Imaris, 3D rendering, microglia, Sholl intersections, convex hull, immune cells, sphericity, chorioamnionitis, intrauterine growth restriction, neonatal hypoxia–ischemia

## Abstract

Microglia morphological studies have been limited to the process of reviewing the most common characteristics of a group of cells to conclude the likelihood of a “pathological” milieu. We have developed an Imaris-software-based analytical pipeline to address selection and operator biases, enabling use of highly reproducible machine-learning algorithms to quantify at single-cell resolution differences between groups. We hypothesized that this analytical pipeline improved our ability to detect subtle yet important differences between groups. Thus, we studied the temporal changes in Iba1^+^ microglia-like cell (MCL) populations in the CA1 between P10–P11 and P18–P19 in response to intrauterine growth restriction (IUGR) at E12.5 in mice, chorioamnionitis (chorio) at E18 in rats and neonatal hypoxia–ischemia (HI) at P10 in mice. Sholl and convex hull analyses differentiate stages of maturation of Iba1^+^ MLCs. At P10–P11, IUGR or HI MLCs were more prominently ‘ameboid’, while chorio MLCs were hyper-ramified compared to sham. At P18–P19, HI MLCs remained persistently ‘ameboid’ to ‘transitional’. Thus, we conclude that this unbiased analytical pipeline, which can be adjusted to other brain cells (i.e., astrocytes), improves sensitivity to detect previously elusive morphological changes known to promote specific inflammatory milieu and lead to worse outcomes and therapeutic responses.

## 1. Introduction

Until recently, microglial morphological studies have been limited to the cumbersome and still subjective process of reviewing the most common characteristics of a group of single cells to conclude the likelihood of a “pathological” cellular milieu. Although these conventional methods are extremely useful in the hands of expert operators, intrinsic selection biases exist, significantly impacting interpretation of experimental images. This is further complicated because, at any given time, all microglia are responding to the surrounding environment; thus, their morphology is evolving, creating a matrix of cells at different stages of transformation to pro-phagocytic and/or pro-inflammatory states and many transitional forms in between [1,2] (Figure 1A,B). Furthermore, maturing microglia morphologically evolve during development, progressing from minimally ramified (‘ameboid’-like) to more classically ‘surveillance’ (steady state) microglia from P2 to P18 in rodents [3,4,5] (Figure 1C).

Fast progress in microstructural image capturing, processing and analysis using machine-learning platforms has been permitted by the enhanced computational capabilities developed in recent years. These platforms enable unbiased morphometric evaluation of tri-dimensionally (3D) reconstructed single cells within a brain section, also enabling characterization of their physical interactions with other cell types and simultaneous data retrieval of all cells within a specific population embedded in any given z-stack [1,6]. Although non-immunofluorescent immunohistochemistry and staining can be used in these analytical platforms using reflection confocal modes [7], use of immunofluorescent staining in whole-thickness (25 to 75 µm) brain sections has been proven to reduce noise-to-signal ratio (NSR) while limiting signal loss using high-quality confocal microscopy [8]. These techniques, combined with modern machine-learning analytical platforms, have promoted novel interpretations of experimental data, shifting paradigms in neuroscience [6,9,10,11].

In our laboratory, we have developed an analytical pipeline using high-magnification and -definition confocal microscopy and Imaris software (Bitplane Oxford Instruments, Belfast, UK)—based algorithms to address the inherent limitations of traditional techniques of neuropathological assessment of developmental brain injury. Use of these highly reproducible machine-learning algorithms across all experimental repeats quantifying at a single-cell resolution differences in cell populations within a z-stack limits selection and operator biases and improves sensitivity to detect morphometric characteristics [6,11,12,13]. Thus, we hypothesized that this reproducible, blinded and unbiased pipeline would improve our ability to detect subtle yet functionally important morphometric differences in microglia between experimental groups comparing various rodent models of perinatal brain injury, specifically two with known significant microglia activation, neonatal hypoxia–ischemia (HI) and chorioamnionitis (chorio) and one traditionally demonstrating subtle findings, intrauterine growth restriction (IUGR). Since Iba1 also detects infiltrating macrophages, we will use the term microglia-like cells (MCLs), as recently proposed by a panel of experts [2].

## 2. Materials and Methods

### 2.1. Animals and Experimental Design

All experiments were performed in strict accordance with protocols approved by the institutional Animal Care and Use Committee (ACUC) at Johns Hopkins University and the University of Utah. Protocols were developed and performed consistent with National Research Council and ARRIVE guidelines.

#### 2.1.1. Intrauterine Growth Restriction Model (IUGR)

Model has been extensively published by us and others previously [11,12,13,14,15,16]. Briefly, pregnant C57BL/6J mice (000664, The Jackson Laboratory, Bar Harbor, ME, USA) were anesthetized at embryonic day (E) 12.5 (12 weeks human gestation) [17] for implantation of a micro-osmotic pump (1007D, 0.5 µL/h, Durect Corporation, Cupertino, CA, USA) infusing either 0.5% ethanol (vehicle, sham) or 4000 ng/µL of U-46619, a thromboxane A2-analog (TXA, Cayman Chemical, Ann Arbor, MI, USA) dissolved in 0.5% ethanol. The pump was implanted retroperitoneally. Dams receiving U-46619 developed maternal hypertension by 24 h after pump implantation, and offspring were on average 15% smaller at birth compared to sham [11,15]. Upon delivery, pups were cross-fostered to unmanipulated dams to minimize complications resulting from experimentation. One male and one female pup from each litter of sham or IUGR group were used for experimentation, and we studied a total of 14 mice for our experiments at P10 and P18.

#### 2.1.2. Chorioamnionitis Model (Chorio)

We used a well-characterized model of chorioamnionitis that recapitulates the responses to transient systemic hypoxia–ischemia (TSHI) and inflammation occurring in humans [18,19,20]. In this model, TSHI and intra-amniotic lipopolysaccharide (LPS) are administrated in pregnant Sprague-Dawley rats (Charles River Laboratories, Wilmington, MA, USA) at E18 (24–26 weeks human gestation) as previously described [21,22,23]. Briefly, uterine arteries are clamped for 60 min, followed by intra-amniotic LPS injection (4 μg/sac) (0111:B4, Sigma, St. Louis, MO, USA). Shams are exposed to anesthesia and laparotomy for 60 min with no arterial clamping or LPS injections. Pups are born at E22 (30–32 weeks in human gestation). We used equal numbers of male and female pups in each experiment from at least 3 different dams per condition. In total, we used 21 pups for our studies at P5–11 and P15–19. P2 naive pups were used solely to demonstrate maturational morphological characteristics (Figure 1C).

#### 2.1.3. Neonatal Hypoxic–Ischemic Model (HI)

Model has been extensively reported previously [24,25,26]. C57BL6 mice littermates were randomized to HI and sham groups at P10 (full-term human equivalent). Pups received isoflurane for anesthesia (3% induction and 1% maintenance) with anesthesia-exposed littermates as control shams. A modified Vannucci model was used with right common carotid artery ligation followed by hypoxia exposure (FiO_2_ = 0.08 at 36 °C for 45 min) after 1 h of rest with dam [24,25,26]. Core body temperatures were monitored with rectal thermocouple microprobe (Ad Instruments, Inc., Colorado Springs, CO, USA). Equal number of male and female pups were used for a total of 21 mice used for our experiments at P11 and P18.

### 2.2. Tissue Preparation

Animals were then perfused through the left ventricle with 10–12 mL of 0.1% heparin in phosphate-buffered saline (PBS) for exsanguination followed by 5 min of 4% paraformaldehyde (PFA) for fixation. Brains were then placed in a 4% PFA for 48 h and then transferred to a 30% sucrose solution for cryoprotection. After tissue achieved full saturation, as assessed by tissue sinking to the bottom of the sucrose solution, brains were placed in dry-ice-cooled 2-methyl butane for snap-freezing. The brains were removed from solution after they achieved white coloration and bubbles ceased to escape from the tissue. Frozen brains were stored in a −80 °C until sectioning using a freezing microtome. The microtome was cooled using dry ice and ethanol. Temperature was maintained throughout the cutting process. All brains were cut into 50 µm thick coronal sections. Tissues sections were placed individually in a 96-well plate containing antifreeze buffer prepare with 16.5 g of sodium acetate in 1 L of distilled water with 20 g of polyvynil pyrolidone (Millipore Sigma, Burlington, MA, USA) and 800 mL of ethylene glycol (Millipore Sigma, Burlington, MA, USA). Plates were stored in a −20 °C freezer until experimentation.

### 2.3. Floating Immunofluorescent Immunohistochemistry

Coronal brain sections containing the anterior (dorsal) hippocampal region were identified. After a 10 min wash in Tris-buffered saline (TBS), sections underwent antigen retrieval using sodium citrate buffer pH 6.0 in an 80 °C oven for 90 min. Tissue was then permeabilized with 0.2% (for P2 and ~P10) or 0.4% (for ~P18) triton X in TBS for 20 min and blocked with 10% normal goat serum (NGS) in TBS-Tween (TBS-T). Sections were then exposed to rabbit polyclonal IgG anti-Iba1 (FUJIFILM Wako, Osaka, Japan, 019-19741; 1:400), which has cross-reactivity with mouse and rat tissue. Sections were incubated in the primary antibody solution containing 4% NGS in TBS-T at 4 °C overnight followed by 2 h incubation in goat anti-rabbit IgG Alexa Flour 568, emitting red fluorescence (Therma Fisher Scientific, Inc, Waltham, MA, A11011; 1:400) in 4% NGS in TBS-T at room temperature. Nuclear DNA was stained with 4′,6-Diamidino-2-Phenylindole, Dihydrochloride (DAPI: Thermo Fisher Scientific, Inc, Waltham, MA, USA, D1306; 1:5000) at 1 µg/mL in TBS. Tissue was washed in TBS, mounted on glass slides after mild drying time and cover-slipped using ProLong™ Glass Antifade Mountant (Thermo Fisher Scientific, Inc, Waltham, MA, USA, P36984). Slides were left to dry upright in a light-protected box overnight, and the edges of each were then sealed.

### 2.4. Image Acquisition

Z-stacks were captured at 1440 × 1440 pixels, 16-bit and averaged X2 using a Plan-Apochromat 63X/1.4 oil DIC M27 objective and 1.0 zoom to produce 101.54 × 101.54 µm uncompressed images from the most dorsal CA1. Two stitched Z-stacks were set to be captured at 1 air unit (0.8 µm slicing) to 568 nm wavelength using a Laser Scanning Confocal Microscope LSM700 AxioObserver (Carl Zeiss AG, Oberkochen, Germany). Protocol specifications were followed in all repeats of each experiment regardless of age of the animal or species; thus, pinhole, gain and offset configurations were reused in all experiments. Detection wavelengths were 300–483 nm for DAPI and 560–600 nm for Alexa 568 (Iba1). Uncompressed czi format was used for processing. Representative 3D reconstructions for all 6 groups at P10–P11 and P18–P19 are shown in Figure 2.

### 2.5. Step-by-Step Protocol for Image Processing Using Imaris Software x64 v9.8.0

#### 2.5.1. Imaris File Converter Software (Appendix A)

Import files using the ‘*Add files*’ button in the left upper quadrant of the screen. Note: remove metadata files (.queue) from the folder prior to conversion (Appendix A). Set output destination (Appendix A).Users may need to set the correct voxel size for pixel classification to prevent image stretching or distortion before proceeding with the conversion.Select Start All; open Imaris software once finished (Appendix A).

#### 2.5.2. Using Imaris x64 v9.8.0

In Arena: click ‘*Observe Folder*’, locate previously determined ‘*Output Location*’ (from A2 step above, Figure 3A1) and ‘*Select Folder*’ (Figure 3A2, example of MBP in deep read channel and Iba1 in red channel)Select a file to enter ‘*Surpass Mode*’ in order to begin image analysis. (Figure 3B)Use ‘*Display Adjustment Window*’ to view ‘*Channel of interest*’ (Figure 3C), ‘*Auto Adjust*’ if necessary to obtain ideal viewing settings (Figure 3D). This does not edit images; rather, it adjusts brightness to better visualize the field for differences in monitors and observer’s preference.

#### 2.5.3. Filament Creation

To begin creating a new render, select the ‘*Filament function*’ (Leaf icon, Figure 4A)

**Step 1.** Setting Preliminary Creation Parameters (Figure 4B)

Click to ‘calculate diameter of filaments from image’ (Figure 4B, red square). If the entire field will not be used, check off the ‘*Region of Interest*’ (ROI) box, where the field of view can be cropped on the x, y or z axis (Figure 4B, green square). Note: another option is that a surface render can be made where automatic creation is skipped and edited manually; a specific ROI can be contoured in drawing mode, copied to different slice positions to create that custom surface and masked to add another channel. This ensures that only specific cells or layers within the custom-contoured region can be rendered and quantified without interference from background noise or measurements from regions of interest (i.e., custom pyramidal cell layer within the CA1).After a render is created, create a protocol by saving the object characteristics using the ‘*Magic Wand*’ icon (Appendix A); tab to access the saved ‘*Favorite Creation Parameters*’ (Figure 4B) is useful to quickly process subsequent files in the same manner.Selecting the ‘*Soma Model*’ checkbox attempts to create an object the actual size and shape of the soma in order to use it as a starting point that is more representative of the cellular volume and area (Figure 4B, yellow square). Limitations with this function are outlined below.

**Step 2.***Determining Process ‘Point Diameters*’ (Figure 4C; note: in the software is named ‘*Dendrite Points Diameters*’). Select the relevant source channel and input ‘*starting*’ and ‘*seed point*’ diameters.

1.Determining Starting Point Diameter

Measurements can be obtained within ‘*Slice Mode*’ (Figure 4D1) by clicking on two separate points marking the diameter of the soma (Figure 4D2, shown as white double head arrows). After recording enough diameters (minimum 3), average them. Note: do not forget the *Z*-axis in scenarios involving oblate structures. This diameter measurement does not need to be exact since starting points will ultimately resize according to immunofluorescence intensity.

2.Determining ‘Seed Point Diameter’

The slice mode measurement process can be repeated for placement of seed points to mark diameter of cellular processes (Figure 4D3). Accuracy is important to allow for proper seed point placement during thresholding; however, this will still be an approximation since the final diameter will be changed according to IF intensity in subsequent steps (Figure 4E).

**Step 3.** Filtering Process Start Points.

Automatic detection is rarely accurate for this step (Figure 5A1), so manual threshold to the highest accuracy in coverage is most likely necessary. Afterwards, holding down shift and right or left clicking to add or delete starting points or seed points, respectively, will allow for refinement (Figure 5A2). Ensure that this point is within the center of the cell by toggling or rotating around various planes of the field since depth placement in the 3D plane may lead to inaccurate calculations (Figure 5A3). In our laboratory, we use DAPI co-staining to ensure that the selected cells have a nucleus within the confinement of the z-stack (Figure 5B). Be cautious with the depth of the starting point during manual insertion.The “Remove Seed Points Around Starting Point” function is important in mitigating false, hair-like filament creation around the higher intensity edges of the soma. This is also a reason why properly determining the starting point diameter is of importance (Figure 5C).The “Remove Disconnected Segments” (seen in White vs. Standard Creation without white) function can also be used to refine filament creation. The maximum gap length can be set in accordance with estimated measured distances between cells. This will keep filament autopathing during creating from making an unnecessary “leap” across the field to connect to a seed point placed in an area of high intensity (Figure 5D).

**Step 4.** Soma Determination.

The ‘Soma Model’ function (Figure 4B, yellow box) may not work well for images that have faint intensity in or around the soma. Attempting to utilize the feature in this scenario will lead to distorted starting points that are not an accurate representation of the cell. It is for this reason using standard spheres set to appropriate diameters allows for more accurate and simplified creation (Appendix A). However, somas could be created by using the colocalization feature if needed. These surface renders would then be added to the filament as starting points, post-creation, in the edit tab using ‘*Import Soma*’ under the ‘*Process Filament*s’ section; how to do this is outlined later.

**Step 5.** Setting Filament Diameter.

If the check box related to this step was not clicked during step one, filaments will be rendered in a 1-pixel default line style. If it was, setting the appropriate diameter allows for additional statistical measurements to be calculated. A good way to reduce subjectivity is by using automatic threshold for diameter of filaments in which the software calculates diameter from image intensity (Appendix A).

#### 2.5.4. Post-Creation Image Processing

For experiments where the images showed excessive background causing discrepancies with seed point thresholding and consequential inaccuracies with filament creation, post-creation image processing or masking may be utilized.

**Step 1.** Negating Interfering Antibody Deposits.

Creating a surface render surrounding these infiltrates can be completed by using a high level of detail (0.1–0.3 μm) and multi-filtering according to intensity, voxel size, sphericity (the ratio of surface area of a sphere of equal volume compared to the surface area of the created render) and oblate ellipticity (deviation from spherical shape due to compression along the diameter). The program allows filters to be stacked in such a way that the only objects being created are those that are then above a certain intensity, below a certain size, more spherical, stretched along different axes or any combination. Since true microglial projections are larger and more linear in nature, surfaces that encapsulate only interfering deposits can be created by filtering for low volume, high intensity and high sphericity. After creation, the voxels within those new surfaces can be masked to zero, creating a new channel that is considerably “cleaner” for more accurate filament rendering (Figure 6A1–A4).

**Step 2.** Reducing Background.

Background subtraction, or one of the other corrective options, is another method to obtain a cleaner image for object creation (Figure 6A5). Use with caution as this can introduce subjectivity and bias to the analysis. In order to prevent bias, research must be blinded to important variables, such as sex, treatment and genotype.

#### 2.5.5. Post-Creation Editing

**Step 1.** Assessment of Creation Accuracy.

In instances where filaments are missing or are falsely created, we use the editing ‘*Pencil*’ icon tab for refining of renders to closely represent the original image (Figure 6B0, red box). Manual creation of filaments can be accomplished by employing the ‘*Autopath*’ method. Using the ‘Paint Brush’ (Figure 6B0, yellow box) icon under the ‘*Draw*’ tab allows for selection of filament end point based on a path of fluorescence intensity based on the original image (Figure 6B1–B3). There are also cases where segment deletion to separate filaments, joining of segments or reassignment of beginning points may be required (Figure 6C1,C2). It is up to the discretion of the person performing this analysis to correct any creation issues potentially involving over- or under-thresholding. It is for those reasons that taking care to figure out the most precise creation parameters possible before creation is critical to limit post-processing manipulation.

**Step 2.** Importing Reconstructed Somas Replacing Starting Points.

Using only Iba1 channel will produce low intensity of signal within the center of the soma (nucleus), not allowing for proper soma detection or surface creation. To overcome this problem, a colocalization channel can be built from Channel A (Ch1-T1: DAPI) and Channel B (Ch2-T2: Iba1) (Figure 7A1, blue and red rectangles). This will create a new channel that combines the Iba1 staining with nucleus location information, as determined with DAPI, to create cell bodies that have confluent intensity throughout, permitting proper thresholding to render cell somas accurately (Figure 7A2). Once a new surface render of these objects has been made, they can be imported to its corresponding filament creation (Figure 7A3).

#### 2.5.6. Sholl Analysis

Sholl intersections are the number of times a cell process in a flattened 2D representation intersects the concentric circles centered in the middle of the cell soma. Once filaments (processes) have been created, Imaris automatically calculates Sholl intersection sum according to the Sholl resolution set within preferences, with the default being 1 µm. This statistic can be exported in the same manner to others within the Vantage plot. To analyze the number of Sholl intersections per cell by radius, individual file data must be exported within the Statistics tab, detailed section, Specific Values drop-down, Filament No. Sholl Intersections and Export Statistics on Tab Display to File icon in the bottom right of the properties menu. Once all the data are saved to one folder, they can be consolidated using an Excel extension tool as outlined below.

#### 2.5.7. Convex Hull Analysis

Instructions to download and link MATLAB software to Imaris can be readily found online (https://www.mathworks.com/products/matlab.html, accessed on 4 January 2023). To summarize, download MATLAB and enable it directly through Imaris by allowing access to the appropriate file path in the “XTension Folders” section within the Custom Tools tab of the main preferences menu. Once setup is completed, convex hull analysis can be applied to microglia renders through image processing, filaments functions and filaments convex hull (Figure 7B).

#### 2.5.8. Data Export

Within the Arena home screen, select all files with completed rendering, select ‘*Vantage Plot*’ in the primary menu bar and find the desired statistical values within the drop-down menu of the ‘*Plot Type*’ section. If a value, such as Filament Volume (sum), is not listed within this menu, open ‘preferences’ > ‘statistics’ and ensure that the check box is filled.View the ‘*Detailed statistics*’ tab and order to detect outliers by visualizing the box and whisker plot. If there is a data point with an area, volume or length measurement of zero, this could be a sign that a lone starting point was falsely placed with no seed points to interconnect with. On the contrary, if a cell has exceedingly large values, it may have improperly conjoined with another cell, and this must be split with a new starting point being accurately placed on that string of filaments. Additionally, some filaments may have been clipped during editing and separated from a starting point. This will provide measurement points within a spreadsheet without assigning to a cell, which could improperly shift columns or rows of data, leading to frameshift mis-assignments and mislabeling errors. Careful assessment of outlier points for rendering errors provides an additional layer of confidence that the rendering is representative of the tissue.Once confident that creation errors are eliminated, click the save button within the ‘Plot Number Areas’ section to export these data to a workable spreadsheet.Within the current version of Imaris, there is no tool to aggregate the Filament number Sholl Intersection points with the specific Sholl sphere radius intersection within Vantage plot. As aforementioned, the detailed specific values data can be accessed within the statistics tab of each file. If all those Excel files are saved to the same folder, an Excel consolidation extension, such as Ablebits (https://www.ablebits.com/downloads/index.php, accessed on 4 January 2023), expedites the process of merging data in multiple spreadsheets to one workbook, selecting all sheets and editing to organize or remove nonessential columns and headers and then consolidating selected worksheets into one sheet. This process places the copied ranges under one another with the names of the source sheets to the first column, which is imperative to match identifiers to their data points. Ultimately, this creates a column layout for each Sholl intersection at every (1 μm) radius for all cells per file.All data from exported files can then be consolidated into a master spreadsheet. Given the large size of the exported files, the file format of .xls should be chosen to avoid data corruption. Consistent file naming allows for new columns to be generated using the FlashFill feature of Word Excel software 6. Pivot tables can be used to summarize data according to different groups. After highlighting the dataset, variables derived from column headers will appear in a list. Variables can be assigned as row, column or filter in order to build out the table.

### 2.6. Statistics

Since normality assumptions were not met, non-parametric Mann–Whitney U test for 2 groups and Kruskal–Wallis non-parametric ANOVA with Dunn–Bonferroni correction for 4 comparisons (among treatments of the same age and among ages of the same treatment) were applied. Results were presented as hybrid box and whisker with vertical dot plots. Data were processed per animal (Figure 8) and per cell (Figure 9, Figure 10 and Figure 11). Although we only included cells with nuclei (DAPI^+^) within the z-stack, to account for the effect of MLCs partially captured within the z-stack due to the proximity to the border, cells with filament length, number of Sholl intersections, sphericity and convex hull volume below the 10th percentile were not included in the final analysis. Significance was assigned by *p*-value ≤ 0.05 in all cases. IBM SPSS Statistics 28v (IBM Corporation, Armonk, NY, USA) was used for analysis.

## 3. Results

### 3.1. Difference in Number of MLCs Reactive to IUGR, Chorio and HI

While local microglia are at increased risk of cell death acutely after injury, distal microglia may respond to the insult by infiltrating to brain regions with worse injury to replace lost cells (replacement), while surviving microglia may self-renew after their acute depletion (repopulation) [27,28]. These processes are specific to the type of fetal/perinatal brain injury, the developmental time at which the injury occurs, the timing of evaluation, the region of interest and the species used for the experiments. Thus, while no differences in the number of Iba1^+^ MLCs were observed at P10 or P18 in mouse IUGR offspring (vs. sham, Figure 8A1) or at P11 and P19 rat chorio pups (vs. sham, Figure 8A2), HI injury at P10 leads to increased numbers of Iba1^+^ MLCs (vs. sham, Figure 8A3). An increased number of Iba1^+^ MLCs was not detected 24 h after HI (P11), perhaps due to acute local microglia loss. In contrast, 8 days after HI (P18), the number of Iba1^+^ MLCs doubles (vs. sham; KW H (3) 11.5; *p* = 0.04 vs. sham, Figure 8A3), probably as a result of mechanisms of replacement, repopulation and macrophage infiltration, as described above [27,28]. Thus, quantification of Iba1^+^ MLCs alone without morphometric evaluation of these cells provides limited information to determine potential functional and biological correlates.

### 3.2. Morphometric Differences in Process (Filament) Length and Volume of Iba1^+^ MLCs between Models of Perinatal Brain Injury

Interpretating the change or lack of change in the number of Iba1^+^ MLCs in the hippocampus for each specific pathological condition is speculative and dynamic as replacement, repopulation and cell death as well as macrophage infiltration factor into these changes. Morphometric measurements of Iba1^+^ MLCs support more refined assessment of the overall physiological state. Larger volumes with few or overly excessive filaments resulting in total filament length at either extreme may suggest reactive microglia (Figure 1A). Unlike fetal models of brain injury (IUGR and chorio; Figure 8B1,B2), postnatal HI resulted in 31.8% shorter filament length 8 days (P18) after the insult (vs. sham; KW H (3) 11.3; *p* = 0.01 vs. sham, Figure 8B3). The lack of differences after HI at P11 may be the result of several factors, including developmentally simplified MLCs in sham, making differentiation from truly ‘ameboid’ reactive MLCs difficult. Trends were similar in the average volume of the MLCs Figure 8C).

### 3.3. Evaluation of Complexity of Iba1^+^ Processes in MLCs in Response to Perinatal Brain Injury Using Imaris Software

Complexity of Iba1^+^ MLC’s projections can be evaluated at the single-cell level by assessing the Sholl intersections as they distance from the soma. Average Sholl intersections per cell per animal can be used to identify differences, as shown in Figure 8D. Similarly, convex hull volume (Figure 8E) and sphericity (not shown) can also be averaged per cell within the z-stack. We strongly believe that morphometric evaluation of Iba^+^ MLCs at single-cell resolution using Imaris-software-derived algorithms improves evaluation of changes between experimental groups not detected with less-refined methods (Figure 8).

#### 3.3.1. Single-Cell Morphometric Analysis of Iba1^+^ MLCs in the IUGR Model

Total filament length was 31.2% shorter in IUGR offspring at P10 (KW H (3) 198.6; *p* < 0.001; *p* < 0.02 vs. sham; Figure 9A); difference not observed by P18, time at which length was 2.2- to 3.5-fold longer than at P10 (*p* < 0.001). Sholl intersections were also decreased in IUGR offspring at P10 (KW H (3) 161.5; *p* < 0.001; *p* = 0.01 vs. sham; Figure 9B) and increased by ~2-fold in both sham and IUGR mice at P18 (*p* < 0.001 vs. P10). Two measurements derived from convex hull analysis, sphericity (Figure 9C) and volume (Figure 9D) did not demonstrate differences between sham and IUGR either at P10 or P18. However, these convex hull measurements demonstrated developmental increase in sphericity (KW H (3) 33.9; *p* < 0.001; *p* < 0.001 P18 vs. P10; Figure 9C) and volume (KW H (3) 190.9; *p*, 0.001; *p* < 0.001 P18 vs. P10; Figure 9D). The decreased filament length and total Sholl intersections were better characterized by analyzing the distribution of intersections away from the soma (Figure 9E). Difference in the distribution was only observed at P10 (*p* < 0.001, Figure 9E,F), suggesting less complexity in CA1 Iba1^+^ MCLs from IUGR mice compared to sham and increased complexity at P18 equivalent to sham. Thus, among the population of Iba1^+^ MLCs in IUGR CA1, the predominant morphology demonstrated decreased complexity at P10 (full-term equivalent), suggesting a more ‘ameboid’ pro-inflammatory and pro-phagocytic phenotype than P10 sham, which resolved by P18.

#### 3.3.2. Single-Cell Morphometric Analysis of Iba1^+^ MLCs in the Chorioamnionitis Model

Total filament length of Iba1^+^ MLCs was similar between sham and chorio groups at P11, but it was decreased by 14.9% (KW H (3) 139.6; *p* < 0.001; *p* = 0.04 vs. sham) at P19. Total length of processes more than doubled between P11 and P19 (*p* < 0.001 P19 sham vs. P11 sham; Figure 10A). Despite the lack of differences in total filament length between sham and chorio groups at P11, the total number of Sholl intersections was increased by 19.8% in the chorio group (KW H (3) 88.18; *p* < 0.001; *p* < 0.001 vs. sham; Figure 10B). As expected, Sholl intersections almost doubled between P11 and P19 (*p* < 0.001). Convex hull analysis of Iba1^+^ MLC in chorio CA1 demonstrated increased sphericity (KW H (3) 133.1; *p* < 0.001; *p* < 0.001 vs. sham; Figure 10C) at P11 and volume (KW H (3) 176.1; *p* < 0.001; *p* = 0.04 vs. sham; Figure 10D) at P19. As demonstrated by the Sholl intersection distribution away from the soma, chorio CA1 Iba1^+^ MCLs were characterized by earlier increased branching at P11 (*p* < 0.001), which appears to resolve by P19 (Figure 10E). Thus, at P11 (*n* = 164) and P19 (*n* = 182), CA1 Iba1^+^ MLCs exposed to chorio demonstrated significant morphometric differences, including hyper-ramification and increased convex hull volume at P11, and transition to a more simplified state at P19, both being characteristic of pro-inflammatory microglial function (Figure 12).

#### 3.3.3. Single-Cell Morphometric Analysis of Iba1^+^ MLCs in the Neonatal HI Model

Next, we evaluated a well-characterized model of perinatal brain injury at a time equivalent to a full-term human newborn. Total filament length was 31% shorter at P11 in Iba1^+^ MLCs from the HI-injured CA1 (KW H (3) 174.6; *p* < 0.001; *p* < 0.001 vs. P11 sham; Figure 11A), although total number of Sholl intersections was modestly increased (KW H (3) 40.3; *p* < 0.001; *p* < 0.001, P11 sham vs. HI; Figure 11B). Although P18 total filament length and Sholl intersections developmentally increased from P11 by 1.4- and 1.3-fold, respectively (*p* < 0.001), these developmental changes are modest in Iba1^+^ MLCs from the HI-injured CA1. As a result, total filament length was 47% shorter at P18 in Iba1^+^ MLCs from the HI-injured CA1 vs. sham (*p* < 0.001). Convex hull analysis found increase in sphericity in Iba1^+^ MLCs from P11 to P18 in both sham and HI groups (KW H (3) 255.4; *p* < 0.001; *p* < 0.001 vs. P11); however, sphericity was higher in Iba1^+^ MLCs from HI-injured CA1 than from sham at both P11 (*p* < 0.006) and P18 (*p* < 0.001, Figure 11C). On the other hand, convex hull volume was decreased in HI Iba1^+^ MLCs by 69% at P11 (KW H (3) 396.8; *p* < 0.001; *p* < 0.001 vs. sham) and by 74% at P18 (*p* < 0.001 vs. sham, Figure 11D). Sholl intersection distribution away from the soma supported that, at P11, the predominant morphology among 350 Iba1^+^ MLCs in the HI-injured CA1 was ‘ameboid’ with short abundant branching (*p* < 0.001, Figure 11E and F1), explaining the increased sphericity and decreased convex hull volume. HI-injured Iba1^+^ CA1 MLC morphology was transitional at P18, with decreased branching in Sholl intersection distribution (Figure 11E) compared to P11; however, it remained with decreased distal branching (*p* < 0.001, *n* = 556). Thus, the predominant morphology among Iba1^+^ MLCs was ‘ameboid’ 24 h after the insult and becomes ‘transitional’ 8 days after insult (Figure 12).

## 4. Discussion

We characterize how use of highly reproducible machine-learning algorithms across all experimental repeats at single-cell resolution enables identification of differences between study groups, attenuating selection and operator biases and improving sensitivity. This analytical pipeline adds to the routine histopathology evaluation and unbiased tri-dimensional quantification of whole-population morphology, including all cells within a z-stack. Here, we use Imaris software to process z-stacks captured using high-quality confocal microscopy to summarize morphometric characteristics of Iba1^+^ MLCs to reveal subtle yet biologically and functionally important differences between reactivity to three fetal/perinatal brain insults, IUGR, chorio and neonatal HI. We show the complex interaction between developmental maturation of MLCs and responses to various brain insults using single-cell morphometric parameters derived from Sholl intersection and convex hull analyses. With these parameters, we differentiate at P10–P11 the morphometric characteristics of Iba1^+^ MLCs, suggesting a predominantly ‘hyper-ramified’ state in rat offspring exposed to chorio at E18 and an ‘ameboid’ state in mouse offspring exposed to IUGR since E12.5. Postnatal HI injury at full-term equivalent (mouse pups at P10) results in ‘ameboid’ and transitional morphology, with significant anomalies as late as 8 days after the injury (2-year-old human equivalent). Knowing that morphological characteristics support specific functional aspects in MLCs [1,29,30], with ‘ameboid’ forms being more pro-prophagocytic and inflammatory and ‘hyper-ramified’ forms being more chronically inflammatory and cytokine-releasing, this type of analysis fosters development of novel hypotheses that may be experimentally targeted. We conclude that this unbiased analytical pipeline, which can be adjusted to other brain cells (i.e., astrocytes), improves the sensitivity to detect previously elusive morphological changes in MLCs, which may promote specific inflammatory and/or phagocytic milieu and lead to worse outcomes and poor response to therapies.

Fast evolution of sophisticated techniques for microstructural image capture, processing and analysis using reproducible and user-friendly software has empowered researchers to quantify tri-dimensional whole-population cellular morphology in an unbiased manner [1,4,6]. Until recently, microglia morphological studies have been limited to the cumbersome and subjective process of reviewing large numbers of cells to conclude the likelihood of a specific “pathological” cellular milieu. In these situations, the operator may impact significantly the interpretation of the images. In this setting, our complementary analytical pipeline using high-quality confocal microscopy and Imaris-software-based techniques addresses these limitations by enabling use of highly reproducible machine-learning algorithms across all experimental repeats to quantify at the single-cell resolution changes within a z-stack to attenuate selection and operator biases. Although quantification of number of MLCs sometimes may provide important insight about the root pathological process, this parameter alone lacks sensitivity or specificity as numbers are heavily dependent on the timing at which other mechanisms of response to injury, named cell death, replacement, repopulation and infiltration, are occurring [27,28,31,32]. These processes are specific to type of fetal/perinatal brain injury, the developmental time at which the injury occurs, the timing of evaluation, the brain region of interest and the animal species.

We show that morphometric evaluation provides a more refined and unbiased assessment of the overall status of the brain region under study. We find Sholl analysis particularly useful to provide unified characterization of all cells embedded in a z-stack. Sholl analysis [33] is a quantitative neurobiological method used to describe the complexity of branching, originally of dendrites. A plot of the number of ‘dendritic’ (processes) intersections versus radial distance from soma describes the Sholl profile, and the integral defining the area under the curve can be used to produce a single measure of complexity [34]. Dendritic complexity, as defined by Sholl analysis, can be used to study neuronal health and associated disease states changes [6,11,12,13,26]. Similarly, convex hull analysis is used to measure the size of the entire cellular field. Traditionally, convex hull analysis compares the area of a polygon, defined as the smallest convex shape that encompasses a set of points [35], containing a skeletonized cell with processes on the 2D plane as compared to a bounding circle. Confocal imaging with accompanying Imaris and MATLAB processing enables a more robust 3D version of convex hull analysis where the tips of distal dendrites create the convex hull construct and associated volume is calculated. Variations in convex hull sizes can be a potential indication of injury, phenotype or restorative transition state, as shown here.

Two fetal/preterm models, IUGR and chorio, were tested to evaluate how our Imaris-derived morphometric pipeline behaves to identify subtle changes at time points remote to the insult. With these methods, we identified at P10–P11 (full-term human equivalent) that i) IUGR results in a predominant ‘ameboid” morphology characterized by decreased total length of the processes and number of Sholl intersections and ii) chorio results in predominantly a ‘hyper-ramified’ bushy state characterized by increased Sholl intersections, sphericity and convex hull volume. In both cases, the functional correlates to the morphology as a pro-inflammatory state, as has been reported by us and others in both models [36,37,38,39]. Additionally, the ‘ameboid’ morphology characterizing Iba1^+^ MLCs in the CA1 IUGR offspring at P10 also supports a pro-phagocytic state [40]. When testing the effects of neonatal HI 24 h after the postnatal insult, we identified a predominant ‘ameboid’ morphology, which, unlike that identified in IUGR Iba1^+^ MLCs, demonstrates increased Sholl intersections, which may be the result of transitional forms. Thus, neonatal HI acutely results in a pro-phagocytic and pro-inflammatory Iba1^+^ MLC morphology, which becomes transitional by 8 days after HI in the mouse hippocampus. These results agree with previous descriptions of the temporal evolution of microglia reactivity after neonatal HI in rodents [25,41,42,43,44].

We have confirmed using three different perinatal brain injury paradigms that our Imaris-based reproducible, blinded and unbiased pipeline improves our ability to detect subtle yet important differences between experimental groups by using a single-cell approach to improve our sensitivity. However, this study is focused only on the hippocampal CA1, is not powered to study the effects of biological sex in the Iba1^+^ MLCs morphology and used only one marker for MLC, Iba1. Nevertheless, this work tested the robustness of the analytical pipeline and was not meant to cover all facets of MLC biology. Correlations with short-term (i.e., cytokines, quinolinic acid production) and long-term outcomes (i.e., behaviors) are warranted to drive functional interpretations of our results.

## 5. Conclusions

Our novel methods enable unbiased study of whole MLC populations at single-cell resolution to characterize the overall response to various perinatal brain insults. We conclude that this unbiased analytical pipeline, which can be adjusted to many other brain cells (i.e., astrocytes), improves sensitivity to detect previously elusive morphological changes, which may promote specific inflammatory milieu and lead to worse outcomes and poor response to therapies.

## Figures and Tables

**Figure 1 life-13-00899-f001:**
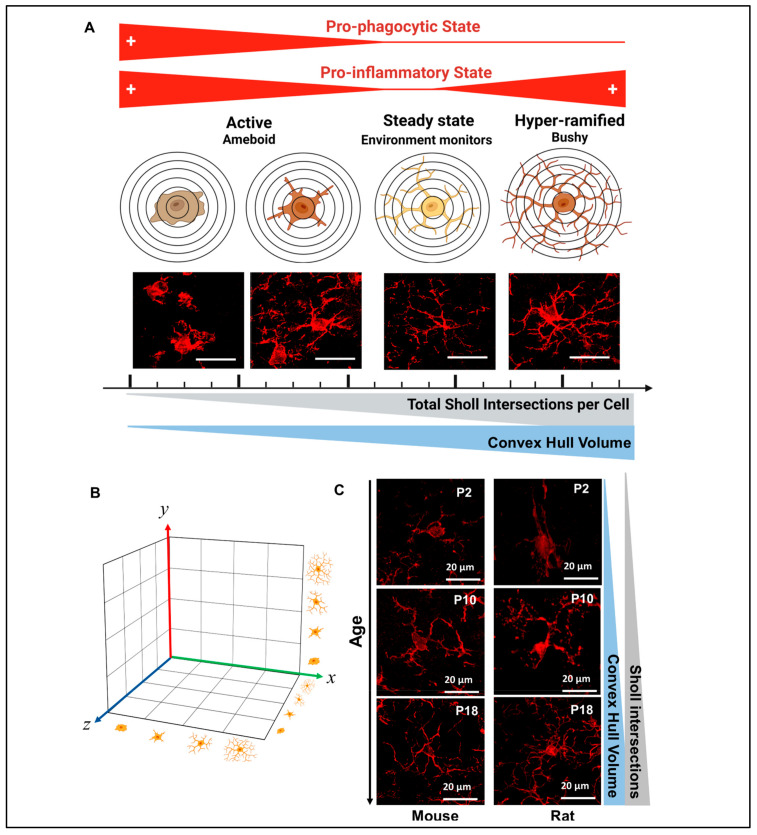
**Morphometric characteristics of Iba1^+^ microglia-like cells (MLCs) depend on developmental stage and pathological condition**. (**A**) Morphometric characteristics are associated with two main functions, phagocytosis and inflammatory. As such, ‘ameboid’-like reactivity state is traditionally linked to pro-phagocytic and pro-inflammatory functions and identified by low number of Sholl intersections and small Convex Hull volumes, while ‘bushy’-like hyper-ramified states are more closely detected in conditions linked to chronic inflammation (i.e., persistent release of pro-inflammatory cytokines) or active pruning and detected with high number of Sholl intersections and large Convex Hull volumes. Bar = 20 µm. (**B**) Although, we attempt to summarize the prominence of specific MLC morphometric state to simplified interpretations, it is important to recognized that each MLC response to their own microenvironment resulting in a matrix of various states of reactivity beyond those 4 depicted in the figure as many transitional states are expected to co-exist. (**C**) Developmentally, MLCs mature from simplified large ‘ameboid’-like cells (P2) to more complex cells with smaller somas acquiring the traditional surveillance state morphology (P18).

**Figure 2 life-13-00899-f002:**
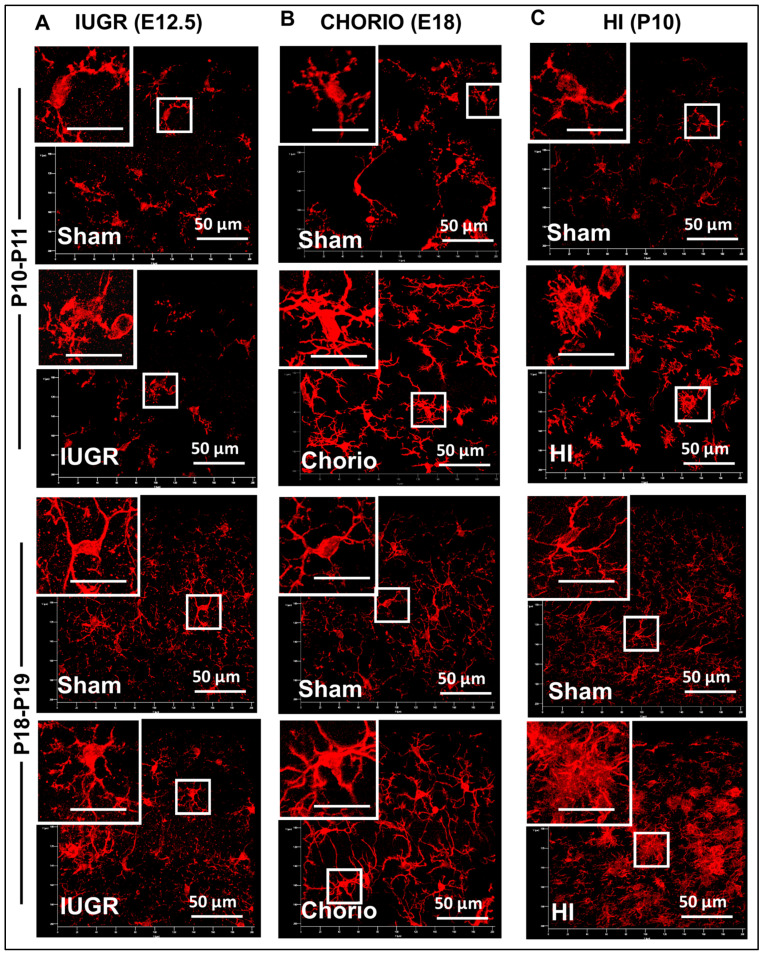
**Microglia-like cell (MLC) responses to various rodent model of developmental brain injury.** Iba1 immunofluorescent floating immunohistochemistry was used in 50 µm-thick slices to identify MLCs in the dorsal hippocampal CA1 at two postnatal time-points, P10-P11 and P18-P19 after: (**A**) intrauterine growth restriction (IUGR) induced by infusion of U-46619, a thromboxane A_2_-analog, during the last week of gestation in mice (E12.5), (**B**) chorioamnionitis (CHORIO) induced by transient hypoxic-ischemic insult for 60 min and LPS injection in rats (E18), and (**C**) neonatal hypoxic-ischemic injury (HI) induced by right carotid artery ligation followed by 45 min of hypoxia at FiO_2_ 0.08 in mouse pups (P10). Characteristic reactivity of MLCs to each of the models of developmental brain injury is shown. Inset bar = 20 µm.

**Figure 3 life-13-00899-f003:**
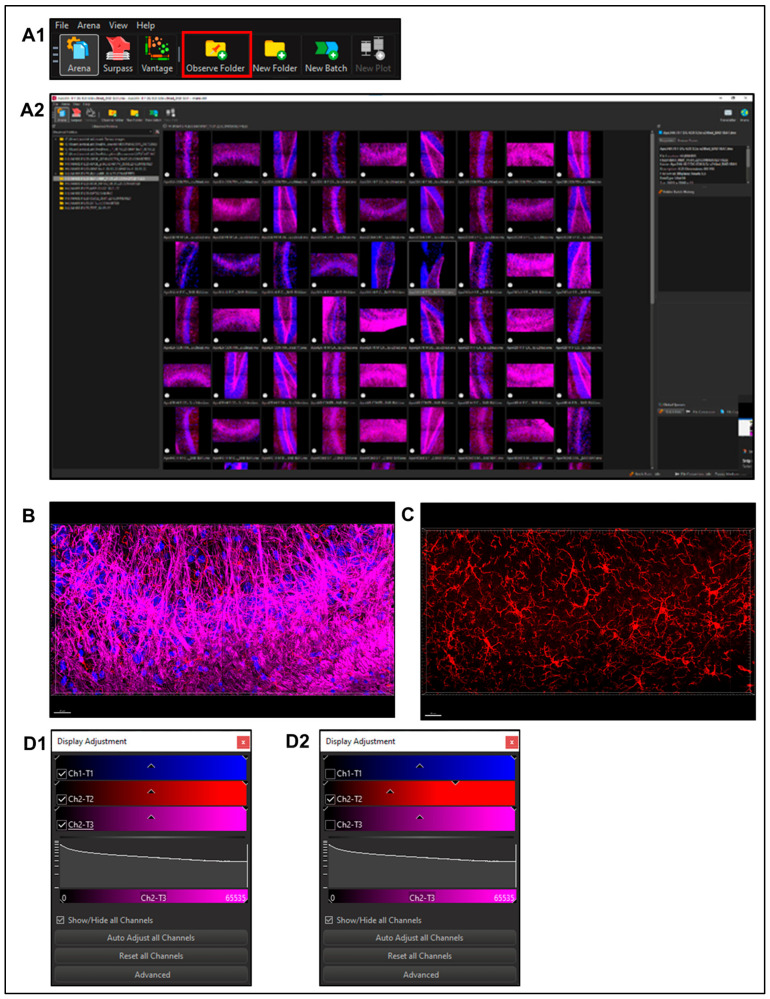
**Using Imaris software to identify the morphometric characteristics of a MLC population.** (**A1**) In Arena: Click ‘*Observe Folder*’, locate previously determined ‘*Output Location*’ and (**A2**) ‘*Select Folder*’ (in the example of MBP in deep read channel and Iba1 in red channel). (**B**) In ‘*Surpass Mode*’ use (**C**) ‘*Display Adjustment Window*’ to view ‘*Channel of interest*’ and (**D1,D2**) ‘Auto Adjust’ if necessary to obtain ideal viewing settings.

**Figure 4 life-13-00899-f004:**
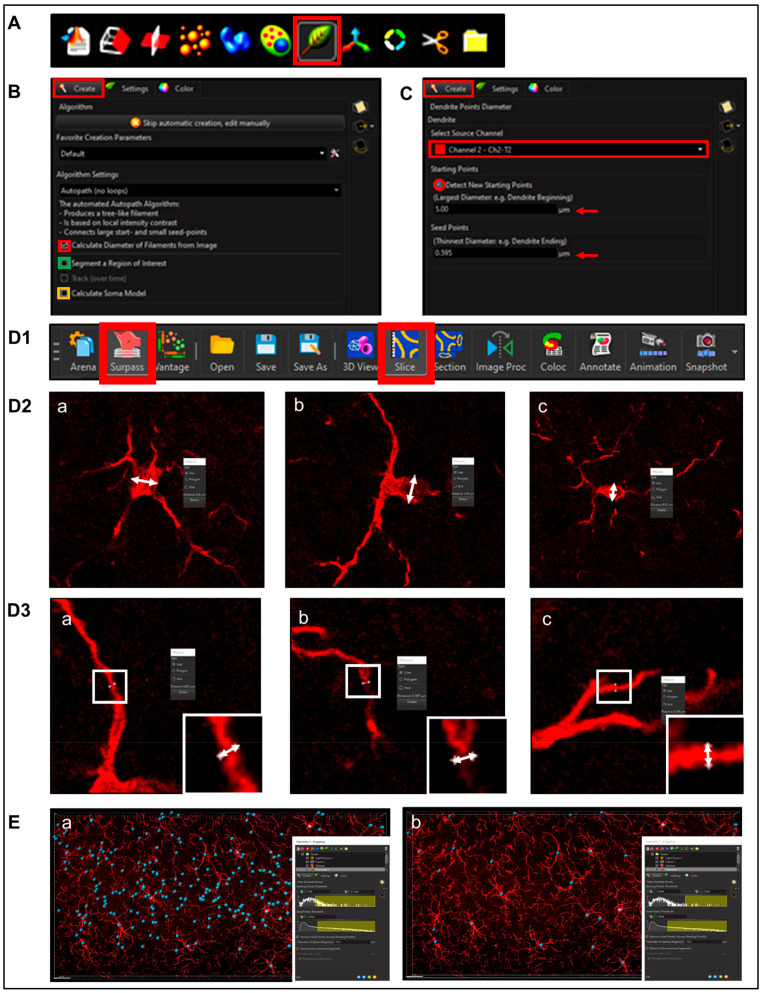
**Filament/process creation.** (**A**) select the ‘*Filament function*’ (Leaf icon, red box) and (**B**) ‘create’ tab then checkbox ‘calculate diameter of filaments from image’ (red square), check off the ‘*Region of Interest*’ (ROI) box If the entire field will be used (green square). Selecting the ‘*Soma Model*’ checkbox attempts to create an object the actual size and shape of the soma in order to use it as a starting point (yellow square). (**C**) Use process ‘*Point Diameters*’ (named ‘*Dendrite Points Diameters’ in software*) and select the relevant Source Channel and input ‘*starting*’ and ‘*seed point*’ diameters obtained in (**D1**) ‘*Surpass*’ and ‘*Slice Mode*’ by (**D2**) clicking on two separate points marking the diameter of the soma (shown as white double head arrows) in triplicate (**a**,**b**,**c**) including the z-axis. (**D3**) Steps are repeated to measure the ‘*Seed Point Diameter*’ to mark diameter of cellular processes. (**E**) The final diameter will be changed according to IF intensity in subsequent steps.

**Figure 5 life-13-00899-f005:**
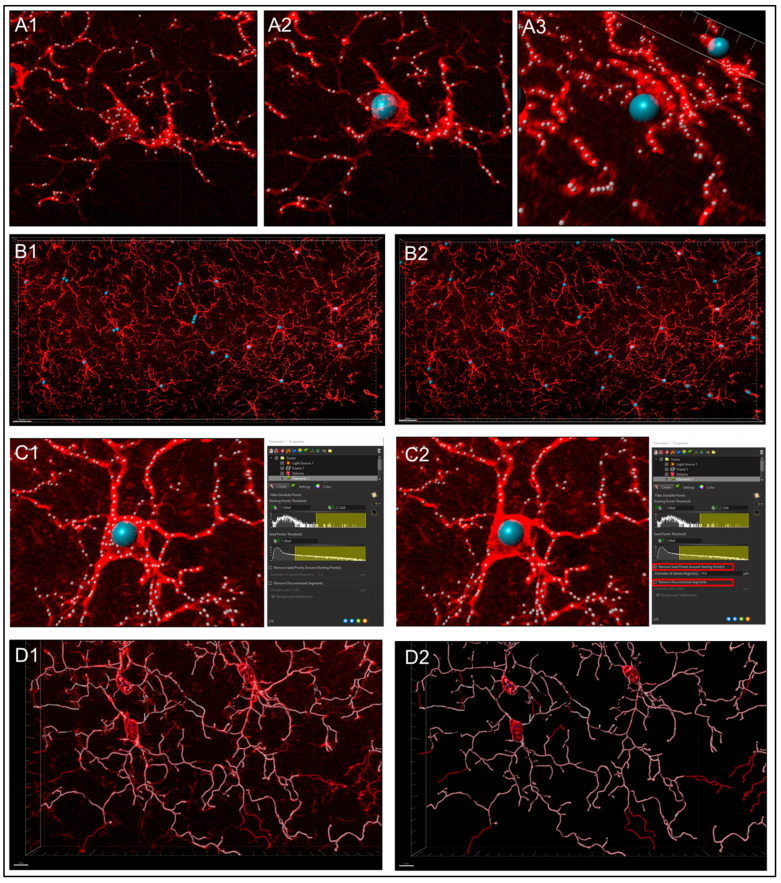
**Filtering process.** (**A1**) Automatic detection is rarely accurate for this step, thus (**A2**) manual threshold to the highest accuracy in coverage is necessary along with adding or deleting starting points or seed points for refinement. (**A3**) Ensure that this point is within the center of the cell by rotating around various planes. (**B1** & **B2**) In our laboratory, we use DAPI co-staining to ensure that the selected cells have a nucleus within the confinement of the z-stack. (**C1** & **C2**) Use ‘Remove Seed Points Around Starting Point’ function to mitigating false, hair-like filament creation around the higher intensity edges of the soma. (**D1** & **D2**) ‘Remove Disconnected Segments’ (Seen in White vs. Standard Creation without white) function can also be used to refine filament creation.

**Figure 6 life-13-00899-f006:**
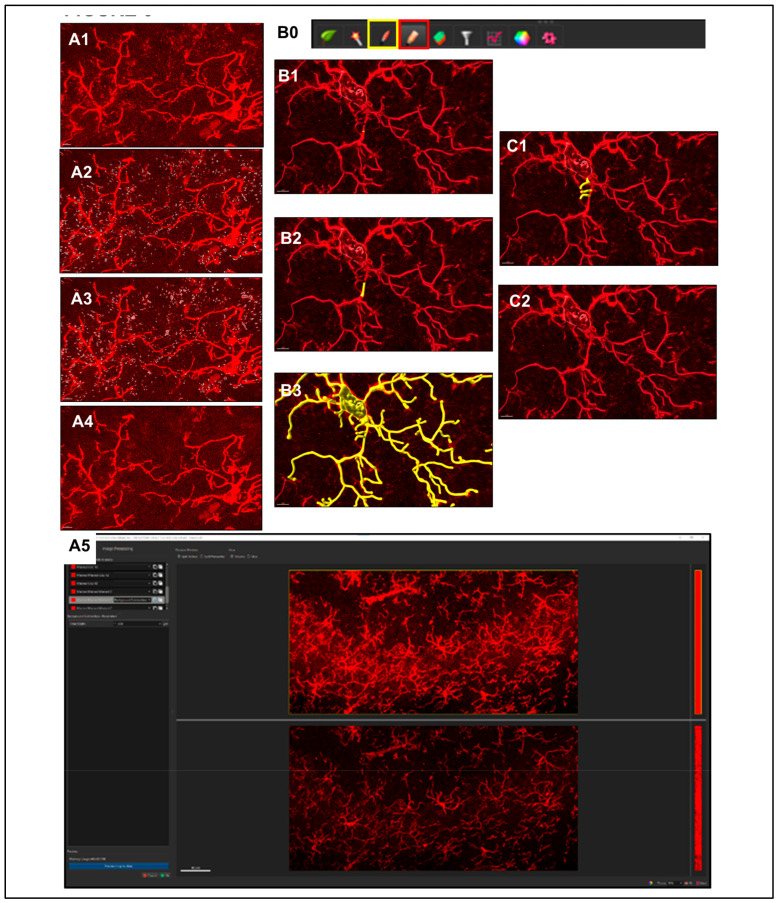
**Post-Creation Image Processing.** (**A1**–**A4**) Since true microglial projections are larger and more linear in nature, surfaces that encapsulate only interfering antibody deposits or debris can be created by filtering for low volume, high intensity, and high sphericity, and after creating mask to zero. (**A5**) Background subtraction another corrective options. (**B**) During post-creation editing accuracy is assessed for missing or falsely created processes. (**B0**) Editing is performed using the ‘*Pencil*’ icon tab (red box) for refining of renders to closely represent the original image. Using the ‘*Paint Brush*’ (yellow box) icon allows for selection of filament end point based on a path of fluorescence intensity based on the original image (**B1**–**B3**). There are also cases where segment deletion to separate filaments, joining of segments, or reassignment of beginning points may be required (**C1**,**C2**).

**Figure 7 life-13-00899-f007:**
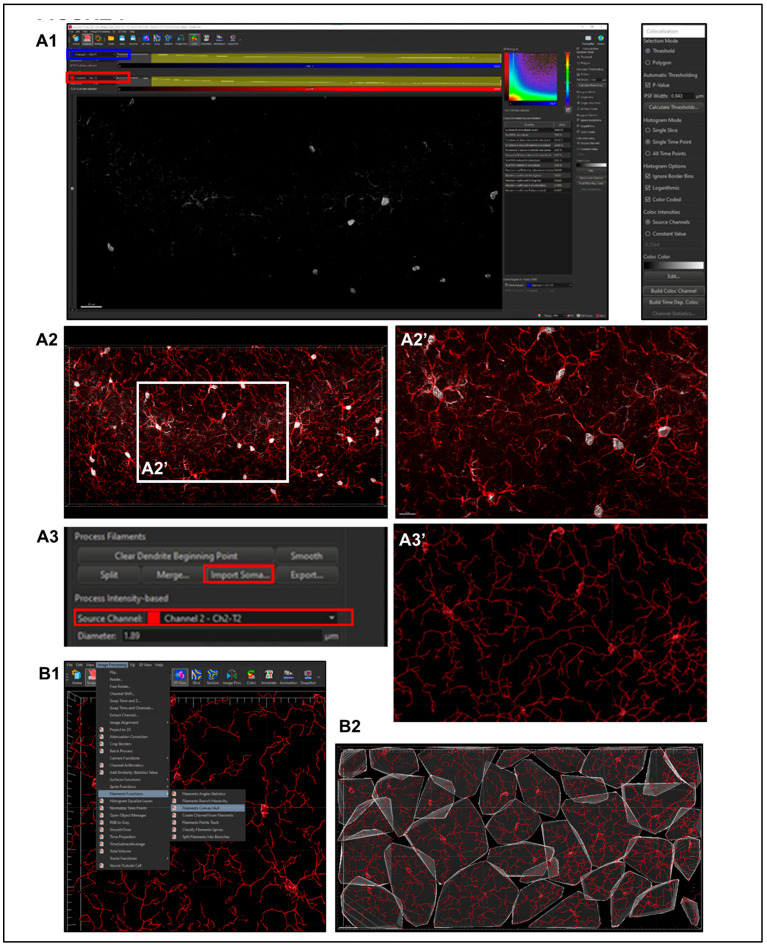
**Importing Reconstructed Somas Replacing Starting Points and Creating Convex Hull Volumes.** (**A1**) Colocalize channels for DAPI (nucleus) and Iba1 (blue and red rectangles). (**A2**,**A2’**) Create a new channel that combines the Iba1 staining with nucleus location information to reconstruct cell bodies permitting proper thresholding to render cell somas accurately. (**A3**,**A3’**) The new surface render can be imported to its corresponding filament creation. (**B1**) Convex Hull Analysis require MATLAB enabled through Imaris. Once setup is completed, the Convex Hull analysis can be applied to microglia renders through Image Processing, Filaments Functions, and Filaments Convex Hull to produce the render shown (**B2**).

**Figure 8 life-13-00899-f008:**
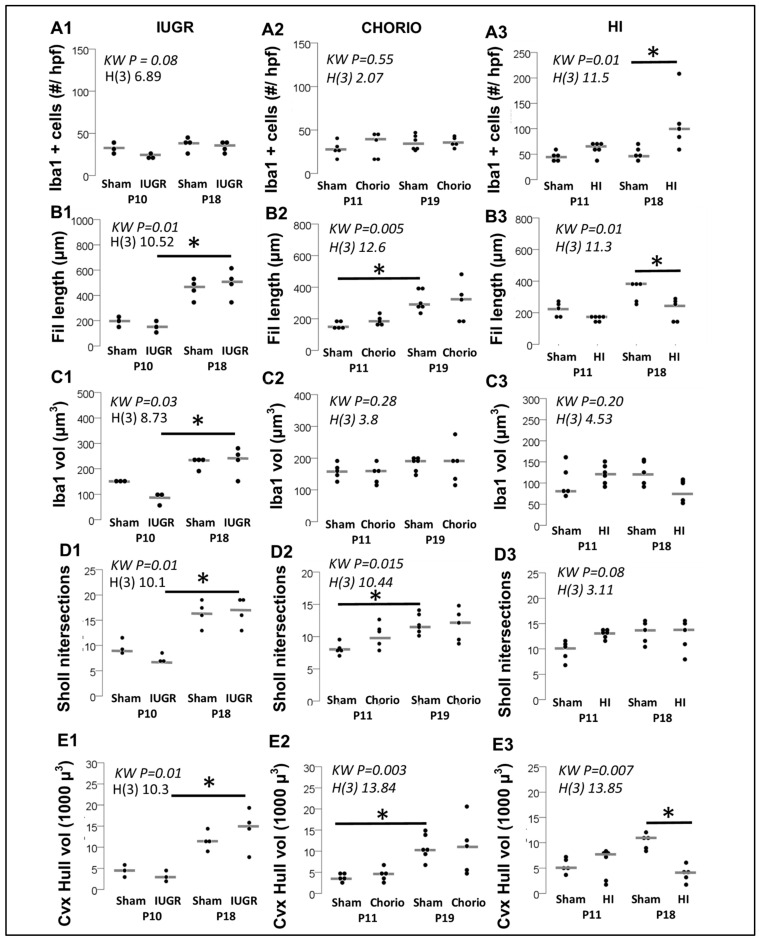
Per animal quantification and morphometric analysis of Iba1^+^ MLC from the CA1 in response to IUGR, Chorio and HI at P10-P11 and P18-P19. (**A**) Iba1^+^ cells (per high-magnification field -z-stack), (**B**) Average filament (process) length (in µm), (**C**) Average volume of Iba1^+^ MLC (in µm^3^), (**D**) Average number of Sholl intersections, and (**E**) Convex Hull volume (in 1000 µm^3^) are represented in a vertical dot plot, with median shown as a grey line. Analysis by Kruskal Wallis ANOVA with Dunn-Bonferroni correction for multiple comparisons. *, *p* < 0.05, n = 5–6 per group.

**Figure 9 life-13-00899-f009:**
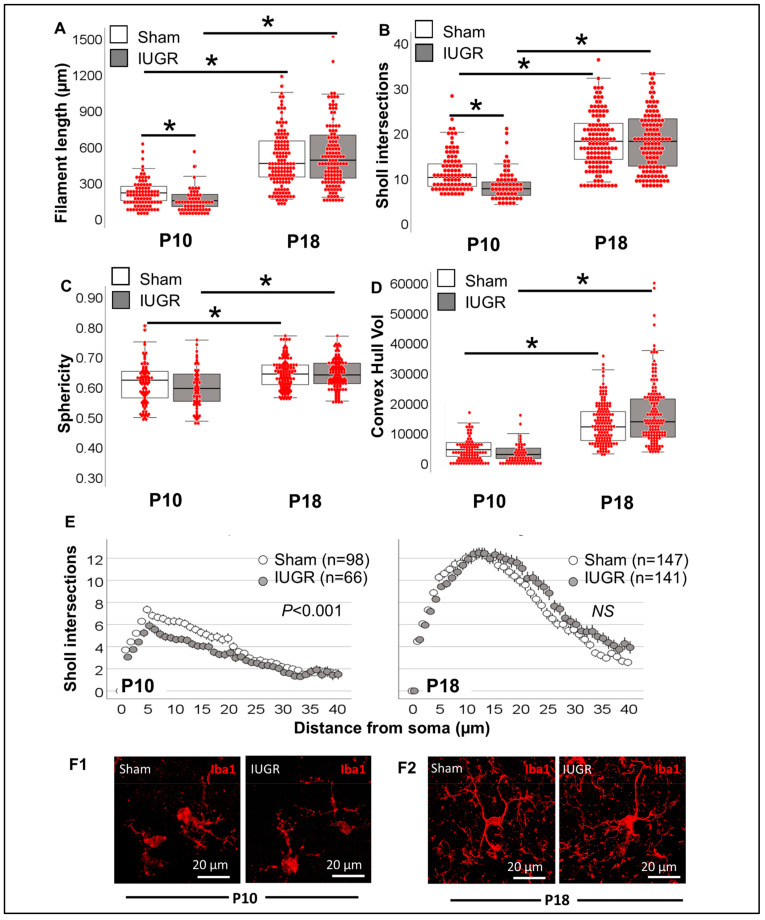
Single-cell resolution analysis improves sensitivity to identify morphometric differences in IUGR and sham Iba1^+^ MLCs. (**A**) Total filament (process) length (in µm), (**B**) total number of Sholl intersections, (**C**) Sphericity and (**D**) Convex Hull volume (in 1000 µm^3^) of sham (white box) and IUGR (grey box) Iba1^+^ MLC are represented in a hybrid box and whisker with vertical dot plot in which each red dot represents a single cell. Analysis was performed by Kruskal Wallis ANOVA with Dunn-Bonferroni correction for multiple comparisons. *, *p* < 0.05. (**E**) The distribution of the number of Sholl intersections as they distance from the soma allows a clear differentiation between groups. Analysis was performed by Kolmogorov-Smirnov test. (**F**) Representative 3D rendering of Iba1^+^ MLCs in sham and IUGR CA1 for P10 (**F1**) and P18 (**F2**).

**Figure 10 life-13-00899-f010:**
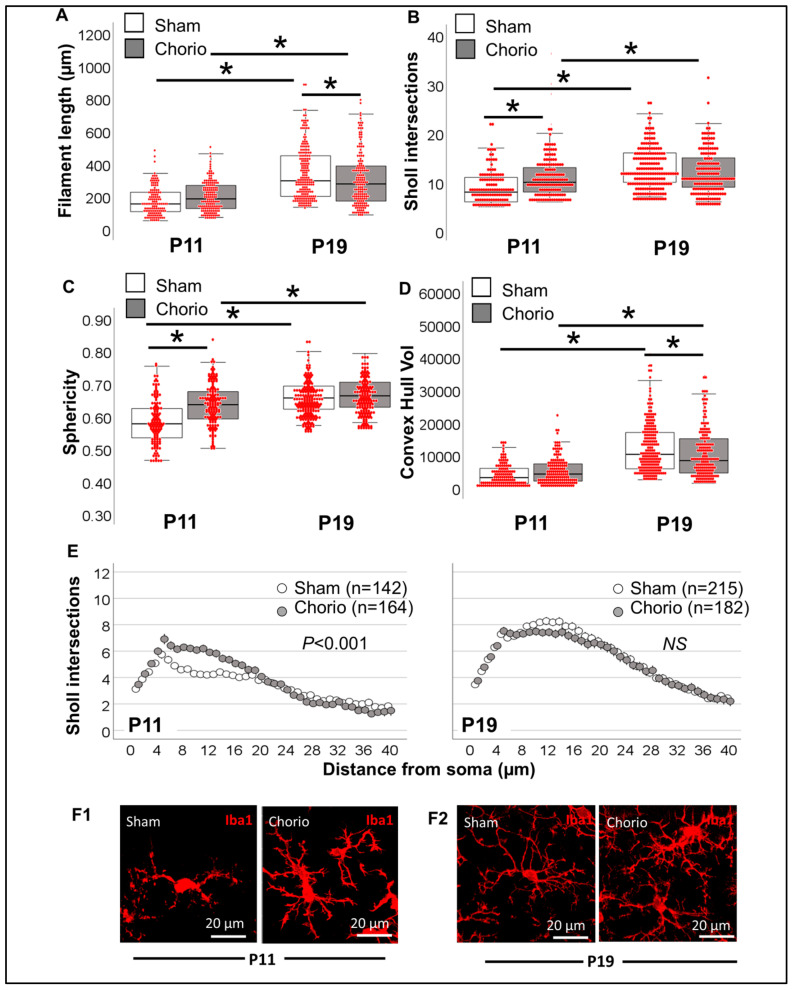
Single-cell resolution analysis improves sensitivity to identify morphometric differences in Chorio and sham Iba1^+^ MLCs. (**A**) Total filament (process) length (in µm), (**B**) total number of Sholl intersections, (**C**) Sphericity and (**D**) Convex Hull volume (in 1000 µm^3^) of sham (white box) and Chorio (grey box) Iba1^+^ MLC are represented in a hybrid box and whisker with vertical dot plot in which each red dot represents a single cell. Analysis was performed by Kruskal Wallis ANOVA with Dunn-Bonferroni correction for multiple comparisons. *, *p* < 0.05. (**E**) The distribution of the number of Sholl intersections as they distance from the soma allows a clear differentiation between groups. Analysis was performed by Kolmogorov-Smirnov test. (**F**) Representative 3D rendering of Iba1^+^ MLCs in sham and Chorio CA1 for P11 (**F1**) and P19 (**F2**).

**Figure 11 life-13-00899-f011:**
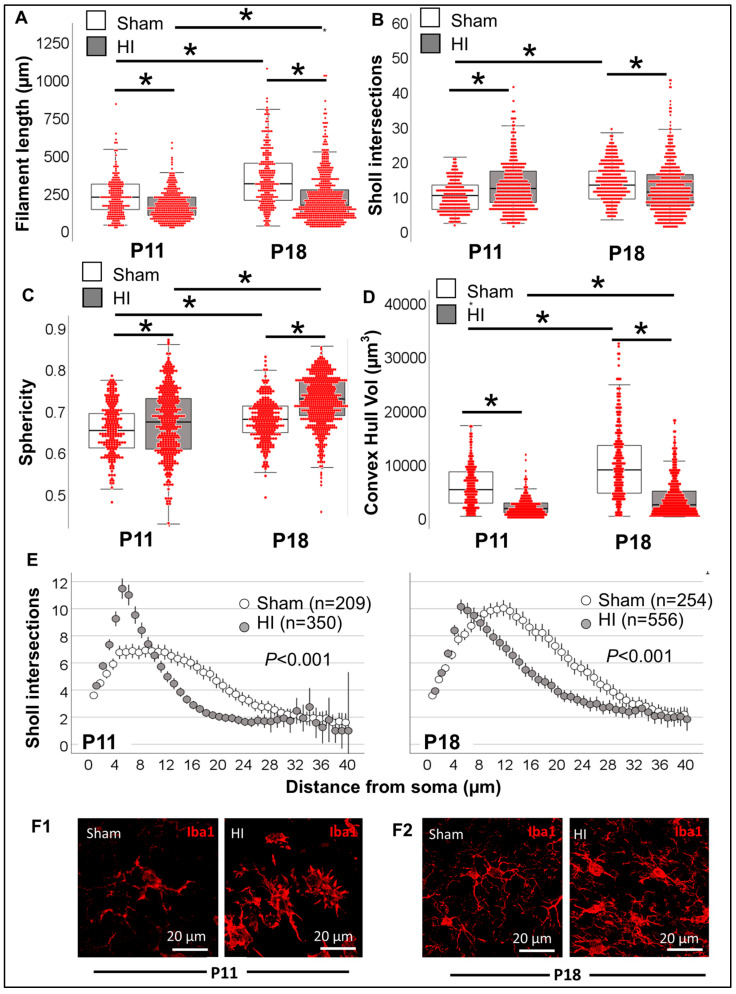
Single-cell resolution analysis improves sensitivity to identify morphometric differences in HI and sham Iba1^+^ MLCs. (**A**) Total filament (process) length (in µm), (**B**) total number of Sholl intersections, (**C**) Sphericity and (**D**) Convex Hull volume (in 1000 µm^3^) of sham (white box) and HI (grey box) Iba1^+^ MLC are represented in a hybrid box and whisker with vertical dot plot in which each red dot represents a single cell. Analysis was performed by Kruskal Wallis ANOVA with Dunn-Bonferroni correction for multiple comparisons. *, *p* < 0.05. (**E**) The distribution of the number of Sholl intersections as they distance from the soma allows a clear differentiation between groups. Analysis was performed by Kolmogorov-Smirnov test. (**F**) Representative 3D rendering of Iba1^+^ MLCs in sham and HI CA1 for P11 (**F1**) and P18 (**F2**).

**Figure 12 life-13-00899-f012:**
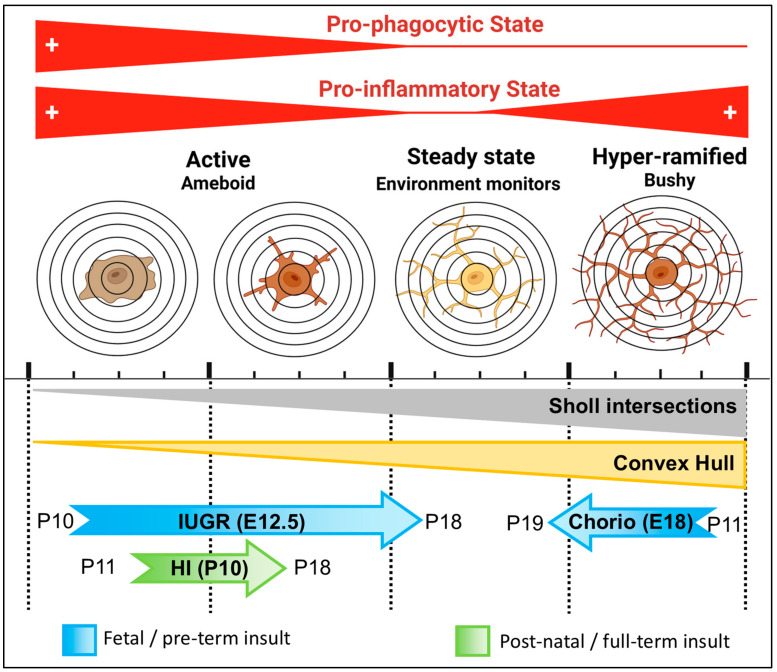
Summary of Iba1^+^ MLC morphometric changes in response to IUGR, Chorio, and HI.

## Data Availability

All data generated or analyzed during this study are included in this article and its Appendix A files. Further enquiries can be directed to the corresponding author.

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
