# Peer review of "Unbiased Quantitative Single-Cell Morphometric Analysis to Identify Microglia Reactivity in Developmental Brain Injury"

_life, 2023, doi:10.3390/life13040899_

Round 1
Reviewer 1 Report
The manuscript entitle “Unbiased Quantitative Single-Cell Morphometric Analysis to 2 Identify Microglia Reactivity in Developmental Brain Injury” describes in detail a new method of analysis of microglia morphology using the IMARIS program, under physiological conditions and under three pathological conditions. The results seem very reliable and the method is well described, in fact it's been pleasurable to see the M&Ms in as much detail as the old-fashioned way. Just a few comments to clarify before the article is a candidate for publication
Minor points:
1. The authors should update in microglial nomenclature in the recent consensus paper of 94 authors published in Neuron 2022. The authors should update their terminology and interpretation of morphological changes with the state of the art in the field. Please see: PMID: 36327895
2. The author should include the antifreeze component buffer in M&M.
3. The author should describe the acronyms of IF (line 218) and change um to µm (line 354).
4. Authors should clarify whether the statistics are per cell or per animal and include it in the figure legend. Perhaps it would be interesting to see the graphs by animal, not by cells studied.
Author Response
1.In the abstract, At P8-P19(?), HI MLCs......, should be P18- 19?
Thank you for this suggestion. We have updated the abstract accordingly (pg. 1, line 37).
2.In the materials and methods, the authors described the step- by-step procedure that the readers could follow. It is well- written and clear. However, the authors should avoid the use of “you”, for example, “you” can then click the save button.....
We have updated the materials and methods section to remove all second person verbiage.
3.In Figure 8A3, the dates P11 and P18 were missing.
We have fixed the label.
4.In Figure 8E1-3, the unit of the Y-axis should be 1000 mm3. /5.In Figures, 9D and 10D, and 11D, the unit of the Y-axis should be mm3.
The unit for Convex Hull volume is µm3.
6.The definition of sholl intersections in Figure 8 seems to be different from Figures, 9D and 10D, and 11D, please describe the definition in the materials and methods clearly.
The definition of Sholl intersection has been added to methods in page 11, line 308-309.
Reviewer 2 Report
The study by St Pierre et al describes morphometric features in microglia at two early stages of postnatal maturation (P10 and P18) after being submitted to three different types of injuries (treatment). Furthermore, the authors describe a step-by-step protocol on how to process confocal microscope photographies to perform the analysis with Imaris and Matlab softwares. The analysis was done in an unsupervised way with algorithms of artificial intelligence. The study is well done, but some minors points need to be addressed or pointed out.
1. The study is done at the single cell level and focusing on microglia morphology, but combination of Iba1 with other microglia specific markers such as TMEM119 would eliminate confounding data from infiltrating macrophages.
2. The study did not describe new microglia morphology and even did not perform functional studies to correlate morphometric data with activation state of the microglia.
3. The authors analyse at the level of single cell the morphometry of microglia submitted to a dynamic process of development, then with a high morphological plasticity. As a result, morphological changes are always observed with age (P10 vs P18).
4. In the case of chorioamnionitis (chorio), it is difficult to find a biological meaning of the morphometric findings.
a) When chorio, some morphological alterations are significant at P11 whereas others are significant at P19 (Chorio vs Sham).
b) Furthermore, Sholl interactions (Fig 10 B) and sphericity (Fig 10C) showed that in sham rats both increased from P11 to P19. That would be the normal process. But treatment (chorio) increased Sholl interactions and sphericity at P11 to normal levels observed at P19. Is chorio inducing an altered morphology at P11 that is a normal morphology at P19?
5. Another surprising data is that in sham mice, sphericity is higher at P18 vs P11 in Fig 9C (as it is also observed in sham rats, Fig 10C) but lower at P18 vs P11 in Fig11C. How reliable is the analysis?
6. In results, page 15 line 385, why the authors stated “acute local microglia loss becoming significant 8 days later (P18)”? I do not observe microglia loss after HI (Fig 8A3).
7. Page 20, line 466. What morphometric analysis the authors are referring to when they state “MLC morphology was transitional at P18 with decreased branching compared to P11”?
8. Figures must show one, two or three asterisks to indicate significant 0.05, 0.01 or 0.001 p-values, respectively.
9. In the statistics paragraph, page 15, could be partially captured cells excluded because of location of the soma was in close proximity to the borders of the z-stack instead of being excluded by having morphometric parameters below the 10th percentile?

Author Response
- The authors should update in microglial nomenclature in the recent consensus paper of 94 authors published in Neuron 2022. The authors should update their terminology and interpretation of morphological changes with the state of the art in the field. Please see: PMID: 36327895
We have used the referenced paper for the nomenclature chosen in the manuscript. Because Iba1 is not specific to microglia and may also detect infiltrating macrophages, the appropriate name is microglia-like cells (Paolicelli et el, Neuron 2022, Page 3463). Since no single cell transcriptomic data exist in these cells, which have studied only by phenomics, we cannot differentiate between ATM, PAM or even disease associate microglia (DAM).
- The author should include the antifreeze component buffer in M&M.
Thank you for this comment. We have added the components of our antifreeze buffer in the following sentence: Tissues sections were placed individually in wells of a 96-well plate containing antifreeze buffer [Sodium Acetate in distilled water with Polyvynil Pyrolidone and Ethylene Glycol]. (Pg.4, lines 140-143)
- The author should describe the acronyms of IF (line 218) and change um to μm (line 354).
We appreciate the attention to detail we have made the changes as requested.
- Authors should clarify whether the statistics are per cell or per animal and include it in the figure legend. Perhaps it would be interesting to see the graphs by animal, not by cells studied.
Thank you for this suggestion. Figure 8 presents the data per animal and figure 9, 10 and 11 present the data per cell.
Reviewer 3 Report
x
The manuscript entitled “Unbiased Quantitative Single-Cell Morphometric Analysis to Identify Microglia Reactivity in Developmental Brain Injury” described the method to measure the morphometric features of microglia collected from P10-P11 and P18-P19 in rodent models developmental insults: intrauterine growth restriction (IUGR) at E12.5 in mice, chorioamnionitis (Chorio) at E18 in rats, and neonatal hypoxia-ischemia (HI) at P10 in mice. The subject of this study is important and experiments were well conducted and suitable for publication.
There are several minor issues.
1. In the abstract, At P8-P19(?), HI MLCs……, should be P18-19?
2. In the materials and methods, the authors described the step-by-step procedure that the readers could follow. It is well-written and clear. However, the authors should avoid the use of “you”, for example, “you” can then click the save button…..
3. In Figure 8A3, the dates P11 and P18 were missing.
4. In Figure 8E1-3, the unit of the Y-axis should be 1000 mm3.
5. In Figures, 9D and 10D, and 11D, the unit of the Y-axis should be mm3.
6. The definition of sholl intersections in Figure 8 seems to be different from Figures, 9D and 10D, and 11D, please describe the definition in the materials and methods clearly.
Author Response
- The study is done at the single cell level and focusing on microglia morphology, but combination of Iba1 with other microglia specific markers such as TMEM119 would eliminate confounding data from infiltrating macrophages. The study did not describe new microglia morphology and even did not perform functional studies to correlate morphometric data with activation state of the microglia. The authors analyze at the level of single cell the morphometry of microglia submitted to a dynamic process of development, then with a high morphological plasticity. As a result, morphological changes are always observed with age (P10 vs P18).
We appreciate the comments. The goal of this paper is to describe a method for morphometric analysis of microglia-like cells using Imaris software and provide and step-by-step protocol. We recognized the limitation of using only Iba1 as a marker, and we do not make any assumption that we are solely studying microglia. This is the reason why we use the term Iba1+ microglia-like cells (Paolicelli et el, Neuron 2022). We used Iba1 as is robustly expressed in the cytoplasm of monocyte-derived microglia-like cells. Unlike Iba1, TMEM119 protein is localized in the membrane making and thus, its detection is less robust particularly for the analysis of distal processes from these cells. Although TMEM119 may be more specific, this marker is not expressed in all microglia and is particularly enriched in adult homeostatic microglia (Paolicelli et el, Neuron 2022). Consistent with this, TMEM119 has not resulted in robust reconstructions for our studies. Validated antibodies are also not available for rats.
- In the case of chorioamnionitis (chorio), it is difficult to find a biological meaning of the morphometric findings.
- a) When chorio, some morphological alterations are significant at P11 whereas others are significant at P19 (Chorio vs Sham).
b) Furthermore, Sholl interactions (Fig 10 B) and sphericity (Fig 10C) showed that in sham rats both increased from P11 to P19. That would be the normal process. But treatment (chorio) increased Sholl interactions and sphericity at P11 to normal levels observed at P19. Is chorio inducing an altered morphology at P11 that is a normal morphology at P19?
As we show in Fig 1 C, Sholl intersections increase as microglia mature from P2 to P18. However, excessive processes for the particular developmental stage results in a hyper-ramified morphology characteristic of pro-inflammatory, cytokine releasing microglia. Compared to sham, chorio microglia-like cells in the hippocampus are hyper-ramified at P11, while they progress to an intermediate stage, closer to a surveillance cell, with advancing developmental stage (As shown in Fig 12)
- Another surprising data is that in sham mice, sphericity is higher at P18 vs P11 in Fig 9C (as it is also observed in sham rats, Fig 10C) but lower at P18 vs P11 in Fig11C. How reliable is the analysis?
We really appreciate the comment from the reviewer as it brought to our attention that the file for P18 HI sphericity data was corrupted after transfer from Imaris. We have revised all panels for figure 11 and an updated version has been created and updated in the results (Page 18, lines 463-466).
6.In results, page 15 line 385, why the authors stated “acute local microglia loss becoming significant 8 days later (P18)”? I do not observe microglia loss after HI (Fig 8A3).
The statement has been changed consistent with this recommendation (Pages 14 and 15, lines 387-391).
- Page 20, line 466. What morphometric analysis the authors are referring to when they state “MLC morphology was transitional at P18 with decreased branching compared to P11”?
The statement has now been clarified on page 19, lines 471-474
- Figures must show one, two or three asterisks to indicate significant 0.05, 0.01 or 0.001 p-values, respectively.
P-values are stated in the text with the rest of the analytical details. Adding more than one asterisk makes the panel busier without providing any additional information to the reader as significance is defined as P <0.05.
- In the statistics paragraph, page 15, could be partially captured cells excluded because of location of the soma was in close proximity to the borders of the z-stack instead of being excluded by having morphometric parameters below the 10th percentile?
Those cells could be excluded manually, but with thousands of cells being analyzed per group this additional step will make the protocol exceedingly labor intensive and also add selection bias. A precise and unbiased definition of “too close” is not available. We have used the 10% percentile to improve rigor and reduce subjective selection bias.